# Serious adverse reactions associated with ivermectin: A systematic pharmacovigilance study in sub-Saharan Africa and in the rest of the World

Jérémy T. Campillo[1,2]*, Michel Boussinesq[1], Sébastien Bertout[1,3], Jean-Luc Faillie[2,4°], Cédric B. Chesnais[1°]

1 TransVIHMI, Université Montpellier, Institut de Recherche pour le Développement (IRD), INSERM, Montpellier, France, 2 Department of medical pharmacology and toxicology, CHU Montpellier, Montpellier, France, 3 Laboratoire de Parasitologie et Mycologie Médicale, Université de Montpellier, Montpellier, France, 4 EA 2415, IDESP, University of Montpellier, Montpellier, France

° These authors contributed equally to this work.
* jeremy.campillo@ird.fr

**Data Availability Statement:** Data cannot be shared publicly because of restriction of access to VigiBase. Data are available from the WHO Global

## Abstract

### Background

Ivermectin is known to cause severe encephalopathies in subjects infected with loiasis, an endemic parasite in Sub-Saharan Africa (SSA). In addition, case reports have described ivermectin-related serious adverse drug reactions (sADRs) such as toxidermias, hepatic and renal disorders. The aim of this study was to identify suspected sADRs reported after ivermectin administration in VigiBase, the World Health Organization's global individual case safety reports database and analyze their frequency relative to the frequency of these events after other antinematodal drugs reported in SSA and other areas of the world (ROW).

### Methods

All antinematodal-related sADRs were extracted from VigiBase. Disproportionality analyses were conducted to investigate nervous, cutaneous, psychiatric, respiratory, renal, hepatic and cardiac suspected sADRs reported after ivermectin and benzimidazole drug administration across the world, in SSA and RoW.

### Principal findings

2041 post-ivermectin or post-benzimidazole suspected sADRs were identified including 667 after ivermectin exposure (208 in SSA and 459 in the RoW). We found an increased reporting for toxidermias, encephalopathies, confusional disorders after ivermectin compared to benzimidazole drug administration. Encephalopathies were not only reported from SSA but also from the RoW (adjusted reporting odds ratios [aROR] 6.30, 95% confidence interval: 2.68–14.8), highlighting the fact these types of sADR occur outside loiasis endemic regions.

Individual Case Safety Report (ICSR) database, VigiBase® for researchers who meet the criteria for access to confidential data. Vigibase has the data we used and makes them available to people working in a hospital pharmacovigilance service via the following link: https://vigilyze.who-umc.org/.

**Funding:** The author(s) received no specific funding for this work.

**Competing interests:** The authors have declared that no competing interests exist.

## Conclusion

We described for the first time suspected sADRs associated with ivermectin exposure according to geographical origin. While our results do not put in question ivermectin's excellent safety profile, they show that as for all drugs, appropriate pharmacovigilance for adverse reactions is indicated.

## Author summary

Ivermectin is a drug used worldwide for various indications: onchocerciasis, lymphatic filariasis, strongyloidiasis, human sarcoptic scabies, acarodermatitis and rosacea. In the early 1990s, it was discovered that ivermectin could induce severe encephalopathies in some patients with high parasite loads of *Loa loa*, a filarial nematode. This objective of this pharmacovigilance study is to summarize serious neurological and non-neurological post-ivermectin adverse drug reactions reported in the World Health Organization database called VigiBase. This study shows that reported serious adverse drug reactions associated with ivermectin are fairly consistent with those mentioned in the official product information of ivermectin but also provides some new signals. Serious post-ivermectin encephalopathies can also occur outside of *Loa loa* endemic regions but the understanding of the mechanism by which it occurs requires further studies. A new signal concerning two serious toxidermias (DRESS syndrome and acute generalized exanthematous pustulosis) is also described. A lack of reporting of adverse drug reactions is noticeable in some Sub-Saharan African countries, and actions are needed to increase the reporting rates of these adverse effects in these countries.

## Introduction

Ivermectin is included in the World Health Organization (WHO) list of essential medicines and is commonly used worldwide. Stromectol (ivermectin 3 mg) and its generics (Arrow Lab, Biogaran, Gerda, Mylan, Pierre Fabre, Sandoz, Zentiva) are mainly distributed in Europe and North America. In Europe, ivermectin is labeled for the treatment of strongyloidiasis, diagnosed or suspected infection with *Wuchereria bancrofti* (the filarial nematode causing lymphatic filariasis) or *O. volvulus* (the filarial nematode causing onchocerciasis), and human sarcoptic scabies. In North America, ivermectin is labeled for the treatment of strongyloidiasis and onchocerciasis. Ivermectin is also used off-label in certain cases of acarodermatitis (skin inflammation due to bites of parasitic mites), rosacea and loiasis (the disease caused by the filarial nematode *Loa loa*) [1,2]. In African countries, ivermectin is distributed at single oral doses of 150–200 μg/kg as part of onchocerciasis and lymphatic filariasis elimination programs (the drug, registered under the name of Mectizan for these indications, is donated by Merck & Co., Inc.). It is used as preventive chemotherapy, i.e. distributed annually (sometimes biannually) using a mass drug administration strategy, i.e. to the entire eligible population of the target communities without individual diagnosis.

Ivermectin is a derivative of avermectins. It acts mainly by binding to the glutamate-dependent chloride channels of invertebrate nerve and muscle cells, causing an increase in membrane permeability leading ultimately to neuromuscular paralysis and death of certain parasites. In subjects with high densities of microfilariae (mf, the larval stages of the filarial

parasites) in the skin or the blood, ivermectin is able to induce complex inflammatory reactions called Mazzotti reactions which include pruritus, rash, fever, malaise, lymphadenopathy, arthralgia, tachycardia, hypotension, edema and abdominal pain [3,4]. These reactions reflect the inflammatory phenomena associated with the destruction of mf by the drug. Since the early 1990, ivermectin has been known to cause potentially fatal encephalopathies in individuals with very high microfilarial density of *L. loa* in the blood (loiasis is endemic only in Central Africa) [5,6], also referred to as "Possible/Probable *L. loa* encephalopathy temporally related to Mectizan" (PLERM). PLERM can occur in subjects with *L. loa* microfilarial density >10,000 mf/mL if measured before treatment or >1,000 mf/mL if measured after treatment [7]. Since then, few studies have been conducted to investigate the frequency of these *Loa*-related adverse drug reactions (ADR) and the mechanisms by which they occur [8].

In 2017, an analysis of the WHO Global individual case safety report (ICSR) database (Vigi-Base) for serious neurological adverse events was conducted [9]. The search identified 52 ivermectin-related ICSRs entered into VigiBase by the pharmacovigilance system of the Democratic Republic of the Congo (DRC) between 2009 and 2013. All patients had central and peripheral nervous system disorders. The mean *L. loa* microfilarial density measured after treatment in these patients was 2149.1 mf/mL, and 61% of them had microfilarial density below 1000 mf/mL, suggesting the possible occurrence of PLERM at low microfilarial density. Another search of the VigiBase was conducted in 2016 to identify serious neurological adverse events other than PLERM after ivermectin administration. The authors found 28 cases of suspected neurological serious ADRs (sADRs) following ivermectin treatment for diseases other than onchocerciasis (10 for scabies, 8 for acarodermatitis, 3 for strongyloidiasis, 5 for lymphatic filariasis, 1 for myiasis and 1 for taeniasis) [10]. This study raised questions about the mechanisms underlying the appearance of these neurological effects. To our knowledge, these studies are the only two that have used a pharmacovigilance database to evaluate the occurrence of post-ivermectin ADRs and they have focused exclusively on neurological events.

Our systematic search of the literature for non-neurological adverse events found 10 cases where ivermectin was associated with cutaneous reactions [11–15], nephropathy [16], psychiatric disorders [17,18], hepatic disorders [19,20] and multiorgan dysfunction syndrome [21]. Clinical trials and observational studies have reported common adverse events such as headache, pruritus, muscle pain, cough, dyspnea, nausea, vomiting, diarrhea, blurred vision, postural hypotension and confusion and more anecdotal effects such as serious skin reactions and edematous swelling [22–24].

In the present study, we searched VigiBase for all the suspected sADRs (not only the neurological ones) reported after ivermectin treatment and after treatment with other antinematodal drugs and conducted disproportionality analyses considering the geographical origin of the reported cases. More specifically, the aims of this study were to identify (i) possible non-neurological pharmacovigilance signals (increased reporting of serious suspected adverse reactions after treatment with ivermectin compared to treatment with other antinematodal drugs), and (ii) possible neurological signals related to indications other than onchocerciasis.

## Methodology

### Data source

Data were extracted from the WHO Global Individual Case Safety Report (ICSR) database, VigiBase [25] which includes more than 20 million cases of suspected ADRs reported by national pharmacovigilance centers in more than 130 countries participating in the WHO Program for International Drug Monitoring [26]. An ICSR is an anonymized report for a single individual who experienced adverse event(s) that may be linked to the use of one or more

drugs. ICSR contains sociodemographic information (age, sex, reporter qualification, country of origin, year of report), information about the drug administration (frequency, dosage, co-medication) and information about the reported adverse event. The latter include the seriousness according to the criteria of the International Council for Harmonisation of Technical Requirements for Pharmaceuticals for Human Use (ICH) [27], adverse event verbatim description and associated terms from the Medical Dictionary for Regulatory Activities (MedDRA) developed by the ICH. All reports of suspected sADRs associated with antinematodal drugs (Anatomical Therapeutic Chemical [ATC] code P02C) from December 2003 (first ever report of ivermectin-associated suspected sADR recorded) up to July 15, 2020 were extracted. Antinematodal drugs included ivermectin, benzimidazole drugs (mebendazole, tiabendazole, albendazole, ciclobendazole, flubendazole, fenbendazole), levamisole, pyrantel, piperazine, diethylcarbamazine, and pyrvinium. Prior to analysis, suspected duplicate reports identified by an automated screening were excluded [28]. When ivermectin had been administered in combination with a benzimidazole or another antinematodal drug, the report was excluded from the analysis. Suspected sADRs were classified following the MedDRA [29], grouped at the System Organ Class (SOC) level and at the individual preferred term (PT) level.

## Study design

We performed disproportionality analyses using the case/non-case method which allows to identify disproportionate reporting, *i.e.* a higher than expected number of adverse reaction reports compared to other reactions recorded in the database by calculating Reporting Odds Ratios (ROR). ROR compares the odds of exposure to ivermectin between cases and non-cases [30,31].

Cases were defined as reports of each suspected sADR of interest identified by a MedDRA PT for severe headache, encephalopathies, confusional disorders, seizures, toxidermias (drug reaction with eosinophilia and systemic symptoms, Stevens-Johnson syndrome, toxic epidermal necrolysis and acute generalized exanthematous pustulosis), psychiatric disorders, suicidal behavior, severe acute respiratory syndrome (SARS), renal disorders, hepatic disorders, cardiac failure, rhythm disorders and Mazzotti reaction. For specific syndromes of interest, we mapped the PTs for the most common symptoms to one variable and used that in the analyses instead of the individual PTs (Table 1).

Non-cases were defined as reports of any other suspected sADR occurring after administration of the same drug.

## Exposure definition

Exposure to ivermectin was identified in the ICSR by the use of ivermectin (ATC code P02CF01) preceding the onset of the serious adverse reaction. Only oral administration of ivermectin was included (topical formulations were excluded).

## Statistical analysis

Descriptive statistics were used to summarize the basic characteristics according to the origin of the ICSR: sub-Saharan Africa (SSA) or the rest of the world (RoW).

Among all suspected sADR reports associated with antinematodal drugs, our primary analyses consisted in calculating the ROR of each suspected sADR of interest (and corresponding 95% confidence interval [95% CI]) for ivermectin compared to benzimidazole drugs using logistic regression models adjusted for age groups, date of the ICSR publication, and origin of the notification (SSA or RoW). The latter can additionally be used as a proxy for ivermectin indication since >99% of subjects with onchocerciasis live in SSA, and ivermectin is usually

**Table 1. Mapping of the PTs for the most common symptoms of syndrome of interest to a new variable.**

| New variable | Algorithm |
|---|---|
| **Encephalopathies** | **At least one of the following PTs** |
| | – Confusion |
| | – Aphasia |
| | – Loss of consciousness |
| | – Coma |
| **Confusional disorders** | **At least one of the following PTs** |
| | – Confusion |
| | – Agitation |
| | – Disorientation |
| **Toxidermias** | **At least one of the following PTs** |
| | – Drug reaction with eosinophilia and systemic symptoms syndrome |
| | – Stevens-Johnson syndrome |
| | – Toxic epidermal necrolysis |
| | – Acute generalized exanthematous pustulosis |
| **Psychiatric disorders** | **At least one of the following PTs** |
| | – Delusion |
| | – Hallucination |
| | – Delirium |
| | – Depersonalization |
| | – Derealization |
| **Renal disorders** | **At least one of the following PTs** |
| | – Renal failure |
| | – Renal impairment |
| | – Renal pain |
| | – Renal injury |
| **Hepatic disorders** | **At least one of the following PTs** |
| | – Hepatitis |
| | – Hepatic failure |
| | – Hepatocellular injury |
| | – Jaundice |
| | – Liver injury |
| | – Hepatic function abnormal |
| **Mazzotti reaction** | **At least two of the following PTs** |
| | – Headache |
| | – Asthenia **or** Fatigue |
| | – Pyrexia **or** Chills |
| | – Arthralgia **or** Myalgia |
| | – Edema **or** Swelling |

not used for lymphatic filariasis control/elimination outside SSA. To explore a potential effect modification by origin, we performed secondary analyses with stratification according to the origin of the notification (SSA and RoW).

Sensitivity analyses were performed using all antinematodal drugs (including benzimidazoles) as the control group instead of benzimidazoles alone and using the same statistical methods.

For all analyses, the p-values in the Tables are indicated by asterisks: ***: p<0.01; **: p≥0.01 to <0.05; *: p≥0.05 to <0.10. For all analyses, "N/A" means that the category is not available or non-applicable.

Analyses were conducted using STATA v.15.1 software (StatCorps, LP, College Station, TX, USA). Maps were created using the mapCountryData package from R statistical software v. 3.5.0.

## Results

### Descriptive analysis of the sADRs reported after treatment with ivermectin

After elimination of duplicates, 2041 suspected sADRs occurring after administration of anti-nematodal agents were reported between December 2003 and July 2020, of which 209 (10.2%) resulted in death. A total of 667 suspected sADRs were reported after ivermectin administration: 208 cases in SSA and 459 in the RoW. Table 2 shows the distribution of cases between SSA and RoW by age, gender, who reported the case, brand name, fatality, reporting period, and indication.

Most cases concerned people aged 18–44 years old (43.7%) and were reported by healthcare professionals (90.0%). Mean age (44.7 ± 22.9 years for all cases) was significantly lower for SSA (32.3 ± 14.6 years) than for RoW cases (51.1 ± 23.8 years). Sex distribution was also significantly different between SSA cases (female:male ratio 1:1.96) and RoW cases (1:1). Stromectol, the most frequently brand name reported in the RoW (62.0%), was not reported at all in SSA. Suspected sADRs were more frequently fatal in the RoW (67 deaths; 14.6%) than in SSA (9 deaths; 4.3%). Onchocerciasis was the most frequently reported indication for ivermectin use and this was particularly the case in SSA. Scabies was the second most frequently reported indication for ivermectin use, all cases being from the RoW (96; 28.0%). The reported SOC are presented in Table 3, the three most reported SOC were "General disorders and administration site conditions" (44.4%), "Nervous system disorders" (31.3%) and "Skin and subcutaneous tissue disorders" (30.4%).

The three countries that reported the highest number of cases were the United States of America (152 ICRS, 22.8%), France (151, 22.6%) and the DRC (115, 17.2%). Distributions by country for the 6 most frequently reported SOC (excluding the SOC "Infections and Infestations" and "Injury, poisoning and procedural complications" for which a causal relationship to drug administration is extremely unlikely) are presented across the world in Fig 1 and across Africa and Europe in Figs 2 and 3, respectively.

The most frequently reported suspected sADRs are presented by SOC in S1A, S1B and S1C Table. The ten most frequently reported suspected sADRs of interest are reported in Table 4.

The syndromes of interest which occurred after ivermectin intake and described in Table 1 are reported in Table 5.

Ivermectin indications for the 23 serious encephalopathies which occurred outside SSA were scabies (8), acarodermatitis (4), strongyloidiasis (4), rosacea (1), onchocerciasis (1) and unknown indications (5). Ivermectin indications for the 32 serious encephalopathies which occurred in SSA were onchocerciasis (30), unspecified filariasis (1) and unknown (1). Indications for ivermectin treatment in cases of serious toxidermia were scabies (9), unknown (12), strongyloidiasis (3), lice (3), acarodermatitis (2), cysticercosis (1), onchocerciasis (1), unspecified filariasis (1) and in one case ivermectin had been administered erroneously. Ivermectin indications for cases of serious Mazzotti reactions were onchocerciasis (28), lice (3), parasitosis (1), strongyloidiasis (1), worms (1), filariasis (1) and not reported (7).

**Table 2. Characteristics of sADRs exposed to ivermectin reported in VigiBase according to geographical origin.**

| Characteristics | Sub-Saharan cases (n = 208) | RoW cases (n = 459) | Total (n = 667) |
|---|---|---|---|
| **Age**, n (%) | | | |
| 0–17 | 23 (11.5%) | 32 (8.3%) | 55 (9.4%) |
| 18–44 | 136 (68.3%) | 119 (31.0%) | 255 (43.7%) |
| 45–64 | 36 (18.0%) | 110 (28.6%) | 146 (25.0%) |
| 65–74 | 3 (1.5%) | 45 (11.7%) | 48 (8.2%) |
| >74 | 1 (0.5%) | 78 (20.3%) | 79 (13.5%) |
| Unknown | 9 | 75 | 84 |
| **Gender**, n (%) | | | |
| Male | 137 (66.2%) | 221 (50.0%) | 358 (55.2%) |
| Female | 70 (33.8%) | 221 (50.0%) | 291 (44.8%) |
| Unknown | 1 | 17 | 18 |
| **Reporter type**, n (%) | | | |
| Healthcare professionals | 185 (94.9%) | 380 (87.8%) | 565 (90.0%) |
| Non-healthcare professionals | 10 (5.1%) | 53 (12.2%) | 63 (10.0%) |
| Unknown | 13 | 26 | 39 |
| **Brand name**, n (%) | | | |
| Stromectol | 0 | 285 (62.0%) | 285 (42.7%) |
| Mectizan | 100 (48.1%) | 8 (1.7%) | 108 (16.2%) |
| Others* | 0 | 45 (9.8%) | 45 (6.7%) |
| Unknown | 108 (51.9%) | 122 (26.6%) | 229 (34.3%) |
| **Fatal**, n (%) | | | |
| Yes | 9 (4.3%) | 67 (14.6%) | 76 (11.4%) |
| No | 199 (95.7%) | 392 (85.4%) | 591 (88.6%) |
| **Reporting period**, n (%) | | | |
| ≤ 2012 | 91 (43.7%) | 86 (18.7%) | 177 (26.5%) |
| 2013–2015 | 33 (15.9%) | 144 (31.4%) | 177 (26.5%) |
| 2016–2018 | 70 (33.6%) | 142 (30.9%) | 212 (31.8%) |
| 2019–2020 | 14 (6.7%) | 87 (18.9%) | 101 (15.1%) |
| **Indications**, n (%) | | | |
| Onchocerciasis | 110 (74.8%) | 10 (2.9%) | 120 (24.5%) |
| Scabies | 0 | 96 (28.0%) | 96 (19.8%) |
| Acarodermatitis | 0 | 80 (23.3%) | 80 (16.3%) |
| Strongyloidiasis | 1 (0.7%) | 64 (18.6%) | 65 (13.3%) |
| Filariasis | 29 (19.7%) | 7 (2.0%) | 36 (7.3%) |
| Rosacea | 0 | 27 (7.9%) | 27 (5.5%) |
| Parasitosis | 1 (0.7%) | 20 (5.8%) | 21 (4.3%) |
| Others** | 6 (4.0%) | 39 (11.4%) | 45 (9.2%) |
| Unknown | 61 | 116 | 177 |

* Soolantra (28), Scabioral (7), Sklice (6), Rosiver (2), Driponin (1), Ivermec (1).

** Error (10), Lice (10), Prophylaxis (6), Skin disease (4), Pruritus (3), Cysticercosis (3), Helminth infection (2), Hookworm (2), Schistosomiasis (2), Loiasis (1), Taenia (1), Worms (1).

## Disproportionality analysis

The results of the disproportionality analyses of sADRs of interest as well as non-cases after administration of ivermectin compared to benzimidazole drugs are presented in Table 6. After

**Table 3. Frequency of reported SOC by regions and in total.** Multiple SOC can be reported in a single ICSR.

| System Organ Class (SOC), n (% of ICSR with mention of the SOC) | Sub-Saharan reports | RoW reports | Total reports |
|---|---|---|---|
| General disorders and administration site conditions | 120 (57.7%) | 176 (38.3%) | 296 (44.4%) |
| Nervous system disorders | 112 (53.8%) | 97 (21.1%) | 209 (31.3%) |
| Skin and subcutaneous tissue disorders | 72 (34.6%) | 131 (28.5%) | 203 (30.4%) |
| Gastrointestinal disorders | 51 (24.5%) | 78 (17.0%) | 129 (19.3%) |
| Infections and infestations | 5 (2.4%) | 77 (16.8%) | 82 (12.3%) |
| Musculoskeletal, connectives tissues disorders | 52 (25.0%) | 26 (5.7%) | 78 (11.7%) |
| Injury, poisoning, procedural complications | 0 | 67 (14.8%) | 67 (10.0%) |
| Psychiatric disorders | 20 (9.6%) | 42 (9.2%) | 62 (9.3%) |
| Respiratory, thoracic, mediastinal disorders | 10 (4.8%) | 52 (11.3%) | 62 (9.3%) |
| Renal and urinal disorders | 28 (13.5%) | 29 (6.3%) | 57 (8.5%) |
| Investigations | 1 (0.5%) | 56 (12.2%) | 57 (8.5%) |
| Eye disorders | 28 (13.5%) | 21 (4.6%) | 49 (7.3%) |
| Hepatobiliary disorders | 1 (0.5%) | 47 (10.2%) | 48 (7.2%) |
| Vascular disorders | 23 (11.1%) | 24 (5.2%) | 47 (7.0%) |
| Blood and lymphatic system disorders | 2 (1.0%) | 42 (9.2%) | 44 (6.6%) |
| Cardiac disorders | 1 (0.5%) | 27 (5.9%) | 28 (4.2%) |
| Metabolism and nutrition disorders | 0 | 24 (5.2%) | 24 (3.6%) |
| Immune system disorders | 2 (1.0%) | 13 (2.8%) | 15 (2.2%) |
| Ear and labyrinth disorders | 6 (2.9%) | 8 (1.7%) | 14 (2.1%) |
| Reproductive system and breast disorders | 6 (2.9%) | 3 (0.7%) | 9 (1.3%) |
| Pregnancy, puerperium, perinatal disorders | 0 | 9 (2.0%) | 9 (1.3%) |
| Neoplasm benign, malignant and unspecified | 0 | 8 (1.7%) | 8 (1.2%) |
| Endocrine disorders | 0 | 7 (1.5%) | 7 (1.0%) |
| Social circumstances | 0 | 4 (0.9%) | 4 (0.6%) |
| Surgical and medical procedures | 0 | 4 (0.9%) | 4 (0.6%) |
| Product issues | 0 | 3 (0.7%) | 3 (0.4%) |
| Congenital, familial and genetic disorders | 0 | 1 (0.2%) | 1 (0.1%) |

adjustment, the relative frequency of serious headaches reported after treatment with ivermectin and with benzimidazole drugs was similar (adjusted ROR [aROR]: 1.22, 95% CI: 0.83–1.78). This was also the case in origin-stratified analysis (aROR: 1.16, 95% CI: 0.68–1.98 and aROR: 1.39, 95% CI: 0.76–2.53 in SSA and RoW, respectively). In contrast, serious encephalopathies were much more frequently reported after ivermectin than benzimidazole treatment, globally (aROR: 9.23, 95% CI: 4.56–18.61), in SSA countries (aROR: 27.1, 95% CI: 6.34–116.1) and in the RoW (aROR: 6.30, 95% CI: 2.68–14.8). Reports of confusional disorders were strongly associated with ivermectin use globally (aROR: 4.05, 95% CI: 1.81–9.09), in the RoW (aROR: 3.66, 95% CI: 1.49–8.87) but not in SSA (aROR: 3.87, 95% CI: 0.76–19.6). Serious seizures were not more frequently reported after ivermectin than after benzimidazole drugs (aROR: 0.49, 95% CI: 0.49–0.97).

Drug Reaction with Eosinophilia and Systemic Symptoms (DRESS) was more frequently reported with ivermectin than with benzimidazole drugs (aROR: 8.59, 95% CI: 1.85–39.9). No adjustments were possible for the analysis of DRESS because of the low number of cases. Serious toxidermias (DRESS, Stevens-Johnson syndrome, toxic epidermal necrolysis and acute generalized exanthematous pustulosis) were more frequently reported with ivermectin than with benzimidazole drugs globally (aROR: 4.43, 95% CI: 2.07–9.47) and in the RoW (aROR: 6.05, 95% CI: 2.76–13.3), but not in SSA countries (aROR: 0.52, 95% CI: 0.05–5.01). It is

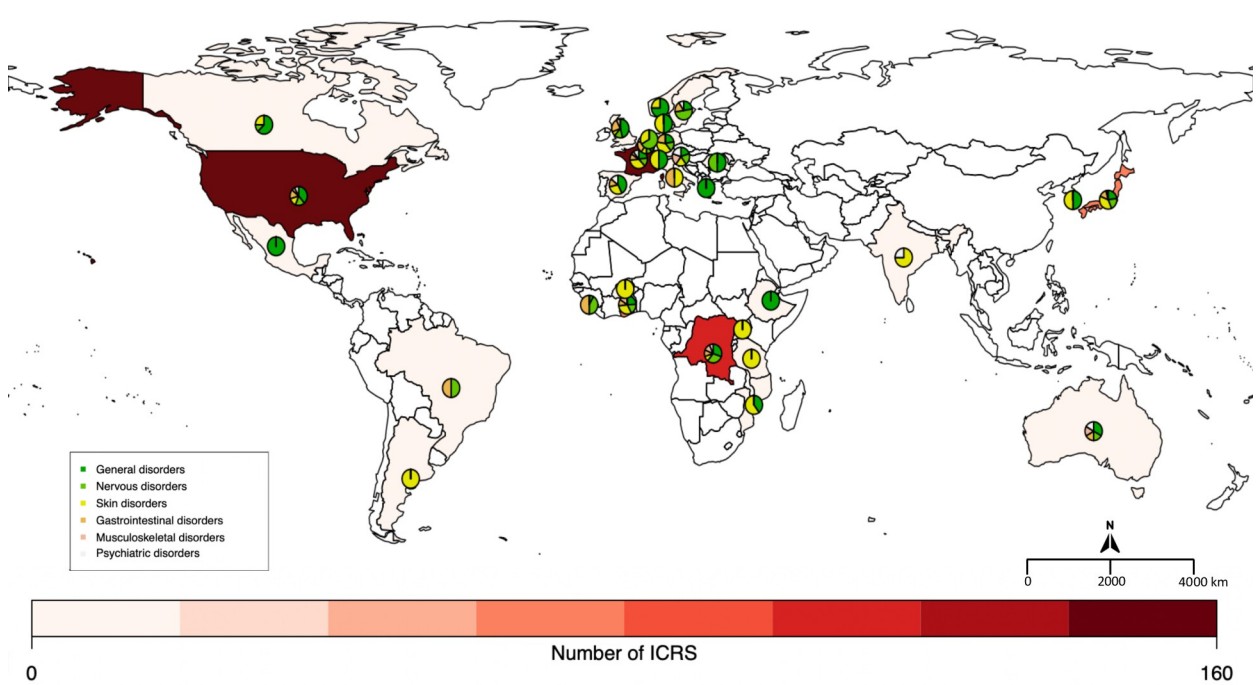

**Fig 1. Number of Individual Case Safety Reports (ICSRs) per country and distribution of sADRs of the 6 most reported SOCs (created with R software and the *Rworldmap* package).** The pie charts show the proportion of the most reported SOCs described by country. The number next to the pie chart represents the number of ICSRs by country (an ICSR can contain multiple SOCs).

noticeable that eight cases of toxidermia were excluded from the analyses because ivermectin was co-administered with albendazole.

Serious psychotic disorders and suicidal disorders were not more frequently reported with ivermectin than with benzimidazole drugs (aROR: 1.78, 95% CI: 0.70–4.53 and aROR: 7.67, 95% CI: 0.85–69.0, respectively).

Only 5 cases of Severe Acute Respiratory Syndrome (SARS) were reported, and no significant associations were found. aROR values did not indicate any associations for serious hepatic disorders or serious renal disorders either (aROR: 0.51, 95% CI: 0.36–0.74 and aROR: 1.36, 95% CI: 0.66–2.85, respectively).

Serious cardiac failures were significantly associated with ivermectin compared to benzimidazole drug intake (ROR: 11.4, 95% CI: 1.37–94.9, no adjustment possible). Serious rhythm disorders were not found to be associated with ivermectin compared to benzimidazole drugs.

Finally, serious Mazzotti reactions were strongly associated with ivermectin compared to benzimidazole drugs both in SSA (aROR: 1.95, 95% CI: 1.09–3.52) and in the RoW (aROR: 19.7, 95% CI: 2.20–175.5).

## Sensitivity analyses

Disproportionality analyses were repeated with all antinematodal drugs rather than only benzimidazole drugs as control group (S2 Table). Associations for reports of serious headache in RoW (aROR: 1.82, 95% CI: 1.01–3.28) and serious rhythm disorders in RoW (aROR: 3.45, 95% CI: 1.02–11.7) were strengthened in these sensitivity analyses. No changes were found for encephalopathies, confusional disorders, DRESS, toxidermias, seizures, renal disorders, suicidal disorders, psychiatric disorders, SARS, hepatic disorders and Mazzotti reactions. In

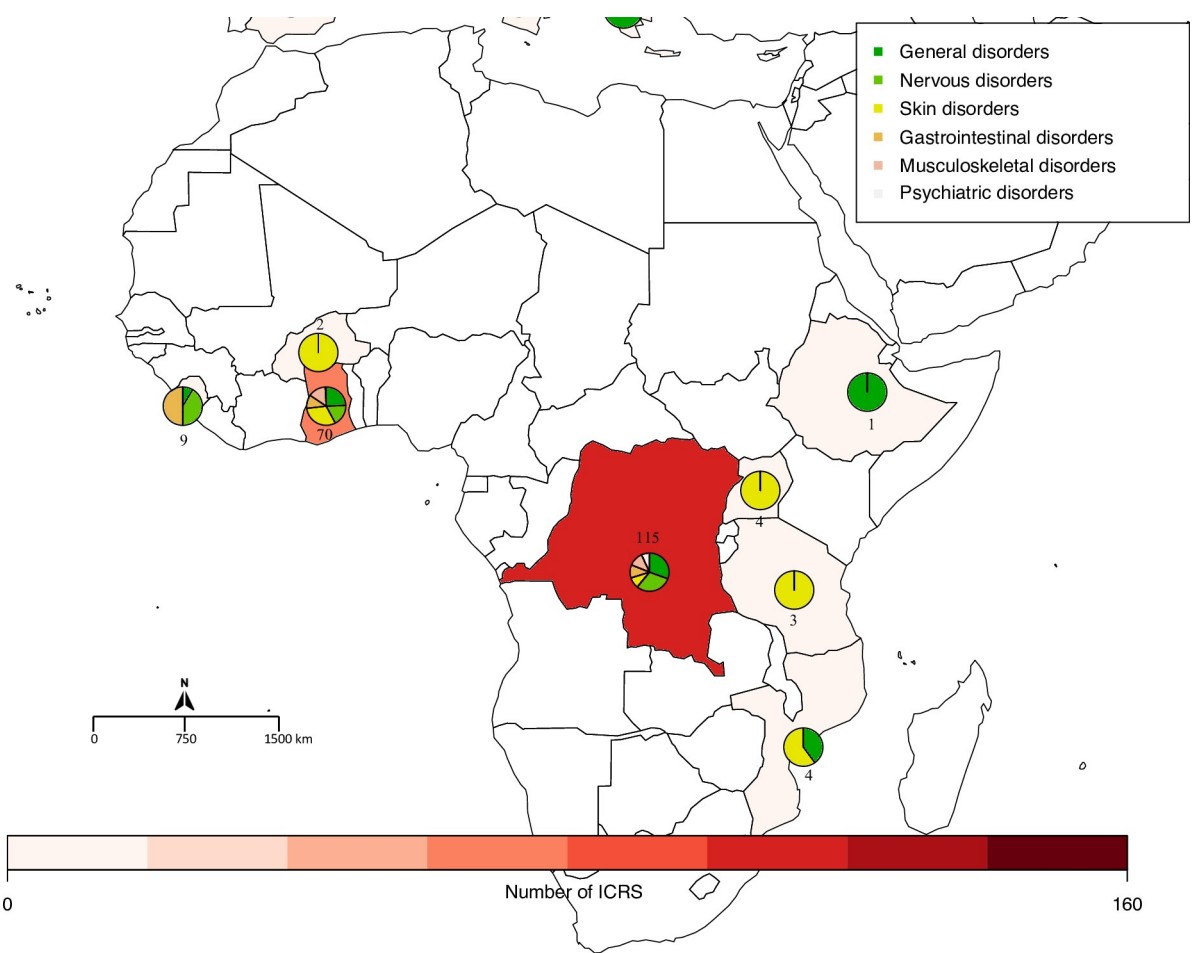

**Fig 2. Number of Individual Case Safety Reports (ICSRs) per African country and distribution of serious ADRs of the 6 most reported SOCs (created with R software and the *Rworldmap* package).** The pie charts show the proportion of the most reported SOCs described by country. The number next to the pie chart represents the number of ICSRs by country (an ICSR can contain multiple SOCs).

contrast to the primary analysis, the sensitivity analysis identified no association for cardiac failures.

## Discussion

Our study used a case-non-case approach to assess the association between the use of ivermectin and the reporting of neurological as well as non-neurological suspected sADRs, recorded in the WHO drug adverse events database from 2003 to 2020 (see S3 Table for STROBE checklist of case-control studies). To our knowledge, it is the first to globally review the main serious ADRs reported with ivermectin. Some strong significant disproportionality signals were found, showing more frequent reporting of encephalopathies after ivermectin than after benzimidazoles, both in SSA countries and in the RoW. Disproportionality signals were also identified for serious toxidermias, serious confusional disorders and serious Mazzotti reactions with ivermectin when compared with benzimidazole drugs or all non-ivermectin antinematodal drugs. A less consistent signal was found for cardiac failures and further studies are needed to confirm this result.

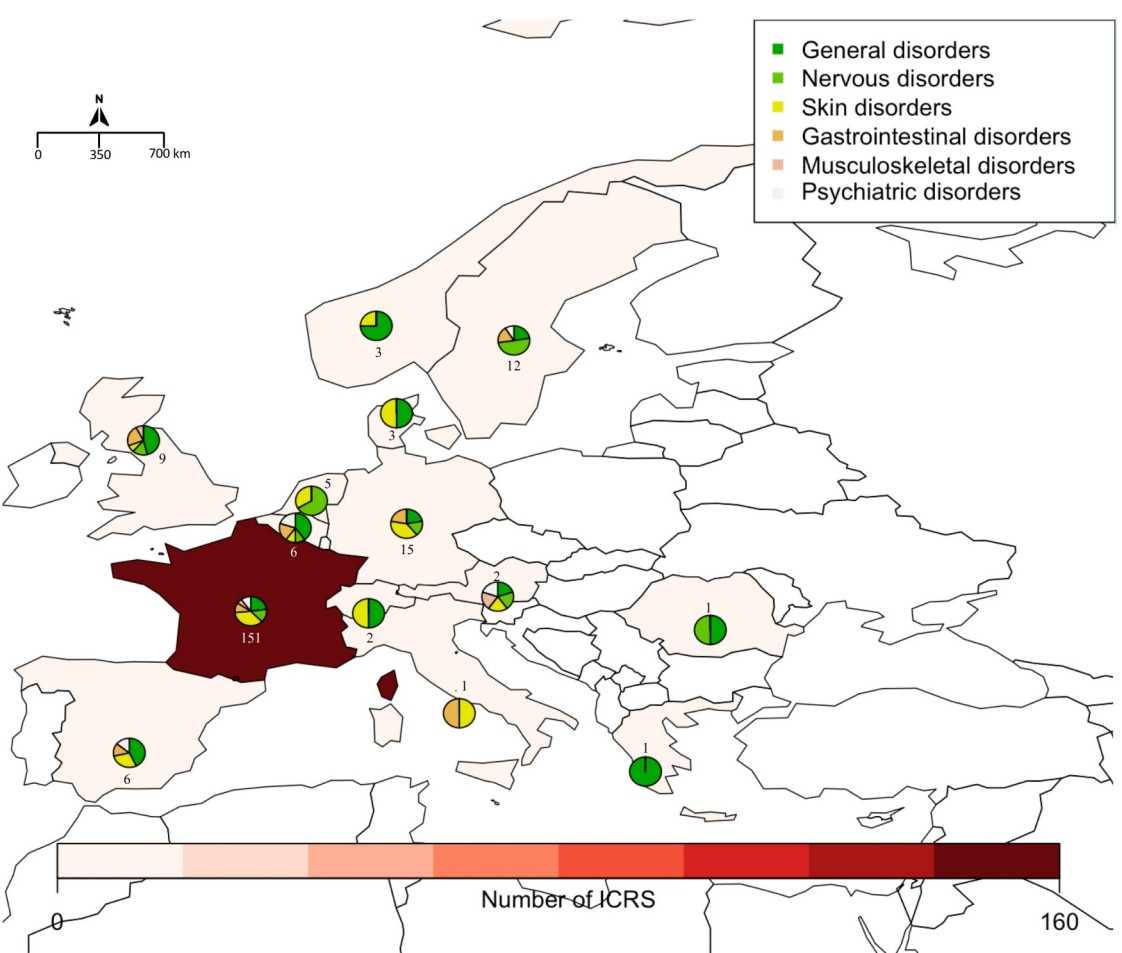

**Fig 3. Number of Individual Case Safety Reports (ICSRs) per European country and distribution of sADRs of the 6 most reported SOCs (created with R software and the *Rworldmap* package).** The pie charts show the proportion of the most reported SOCs described by country. The number next to the pie chart represents the number of ICSRs by country (an ICSR can contain multiple SOCs).

Some of these results were expected, given the different mechanisms of action of ivermectin and benzimidazoles on the various targeted parasites. Ivermectin exerts a strong microfilaricidal effect on filariae, leading to a destruction of mf within one week after treatment. In subjects

**Table 4. Frequency of ten most reported suspected sADRs by regions and in total.**

| Suspected sADRs, n (%) | Sub-Saharan sADRs | RoW sADRs | Total sADRs |
|---|---|---|---|
| Headaches | 60 (73.2%) | 22 (26.8%) | 82 |
| Asthenia | 61 (78.2%) | 17 (21.8%) | 78 |
| Pruritus | 47 (61.8%) | 29 (38.2%) | 76 |
| Pyrexia | 42 (66.7%) | 21 (33.3%) | 63 |
| Coma | 31 (86.1%) | 5 (13.9%) | 36 |
| Dizziness | 29 (67.4%) | 14 (32.6%) | 43 |
| Vomiting | 16 (50.0%) | 16 (50.0%) | 34 |
| Rash | 9 (29.0%) | 22 (71.0%) | 31 |
| Diarrhea | 20 (66.7%) | 10 (33.3%) | 30 |

**Table 5.  Frequency of reported suspected syndromes by regions and in total.**

| Suspected syndromes, n (%) | Sub-Saharan cases | RoW cases | Total cases |
|---|---|---|---|
| Encephalopathy | 32 (58.2%) | 23 (41.8%) | 55 |
| Confusional disorders | 6 (27.3%) | 16 (72.7%) | 22 |
| Toxidermia | 8 (24.2%) | 25 (75.8%) | 33 |
| Psychotic disorders | 1 (9.1%) | 10 (90.9%) | 11 |
| Renal disorders | 1 (5.3%) | 18 (94.7%) | 19 |
| Hepatic disorders | 2 (3.8%) | 51 (96.2%) | 53 |
| Mazzotti reaction | 34 (81.0%) | 8 (19.0%) | 42 |

infected with *Onchocerca volvulus*, the destruction of mf in the skin is associated with inflammatory processes leading to the so-called Mazzotti reaction. In those infected with *L. loa*, the drug probably induces a paralysis of the *L. loa* mf, which are then drained passively in the blood circulation. If the microfilarial density is high, the process can lead to an embolization of mf in the brain capillaries, to inflammatory reactions at the cerebral level, and to an encephalopathy. In contrast, benzimidazoles have little short-term effect on the mf of any filarial species, and thus do not induce Mazzotti reactions, but impair the production of new mf by the adult female worms.

The US Food and Drug Administration (FDA) approved product information for ivermectin mentions that "Rarely, patients with onchocerciasis who are also heavily infected with *Loa loa* may develop a serious or even fatal encephalopathy either spontaneously or following treatment with an effective microfilaricide." [2] In our study, we confirmed the findings of a previous analysis of the data in VigiBase [10] which identified encephalopathies reported with ivermectin also outside of SSA where *L. loa* is not endemic. In addition, we quantified this association by estimating aRORs for ivermectin-induced encephalopathy. aROR was higher in the SSA countries than in the RoW but both were significant, demonstrating a strong global safety signal. Another recent publication described the case of a 13 years old boy presenting a progressive encephalopathy after a single oral dose of ivermectin given at 230 μg per kg, i.e. only slightly higher than the dose used for ivermectin mass drug administration for onchocerciasis (150 μg/kg) and lymphatic filariasis (200 μg/kg) control or to prevent scabies infection (200 μg/kg). The authors found that the patient was a carrier of non-sense mutations in the gene coding for the ATP-binding cassette subfamily B member 1 (ABCB1) transporter which is known to efflux ivermectin from the brain. These mutations can lead to neurological adverse reactions induced by ivermectin [32]. Our results are therefore consistent with the literature and support the evidence of post-ivermectin serious neurological ADRs in some people not infected with *L. loa*. The clinical presentations of *Loa*-related and non-*Loa*-related post-ivermectin neurological ADRs are summarized in Table 7.

The FDA-approved product information for ivermectin also includes the risk of toxic epidermal necrolysis and Stevens-Johnson syndrome as very rare events. By identifying a strong disproportionality signal between ivermectin and benzimidazole drugs or all other antinematodal drugs, our study suggests that ivermectin may be associated with a higher risk of toxidermias than other antinematodal drugs. We also found in VigiBase two types of toxidermia that were never mentioned in the literature: 9 cases of DRESS and 2 cases of acute generalized exanthematous pustulosis after ivermectin intake. These findings could be of great interest for clinicians considering ivermectin treatment in patients at risk for these sADRs or assessing the causality of ivermectin in the development of a toxidermia.

Our study has several strengths. First, we used the global ADRs database VigiBase intended to collect information on suspected ADRs from nearly all national pharmacovigilance centers

**Table 6. Disproportionality analysis of serious adverse reactions associated with ivermectin compared to benzimidazole drugs.**

| Serious adverse drug reaction | Drugs | Cases | Non-cases | Crude ROR (95% CI) | Adjusted ROR [a] (95% CI) | Adjusted ROR [b] in Sub-Saharan Africa (95% CI) | Adjusted ROR [b] in RoW (95% CI) |
|---|---|---|---|---|---|---|---|
| Headache | **Ivermectin** | 71 | 502 | 1.40 ** (1.01–1.93) | 1.22 (0.83–1.78) | 1.16 (0.68–1.98) | 1.39 (0.76–2.53) |
| | Benzimidazoles | 99 | 980 | Ref. | Ref. | Ref. | Ref. |
| Encephalopathy | **Ivermectin** | 55 | 518 | 10.3 *** (5.35–19.9) | 9.23 *** (4.56–18.6) | 27.1 *** (6.34–116.1) | 6.30 *** (2.68–14.8) |
| | Benzimidazoles | 11 | 1068 | Ref. | Ref. | Ref. | Ref. |
| Confusional disorders | **Ivermectin** | 22 | 551 | 3.88 *** (1.87–8.05) | 4.05 *** (1.81–9.09) | 3.87 (0.76–19.6) | 3.66 *** (1.49–8.97) |
| | Benzimidazoles | 11 | 1068 | Ref. | Ref. | Ref. | Ref. |
| Seizure | **Ivermectin** | 11 | 562 | 0.49 (0.25–0.97) | 0.83 (0.40–1.72) | N/A | N/A |
| | Benzimidazoles | 41 | 1038 | Ref. | Ref. | | |
| DRESS * | **Ivermectin** | 9 | 564 | 8.59 *** (1.85–39.9) | N/A | N/A | N/A |
| | Benzimidazoles | 2 | 1072 | Ref. | | | |
| Toxidermia | **Ivermectin** | 25 | 548 | 3.47 *** (1.79–6.73) | 4.43 *** (2.07–9.47) | 0.52 (0.05–5.01) | 6.05*** (2.76–13.3) |
| | Benzimidazoles | 15 | 1065 | Ref. | Ref. | Ref. | Ref. |
| Psychotic disorders | **Ivermectin** | 10 | 563 | 1.72 (0.73–4.08) | 1.78 (0.70–4.53) | N/A | 1.62 (0.62–4.23) |
| | Benzimidazoles | 11 | 1068 | Ref. | Ref. | | Ref. |
| Suicidal behavior | **Ivermectin** | 4 | 569 | 7.58 * (0.84–68.0) | 7.67 * (0.85–69.0) | N/A | N/A |
| | Benzimidazoles | 1 | 1078 | Ref. | Ref. | | |
| SARS ** | **Ivermectin** | 4 | 569 | 7.58 * (0.84–68.0) | N/A | N/A | N/A |
| | Benzimidazoles | 1 | 1078 | Ref. | | | |
| Renal disorders | **Ivermectin** | 17 | 556 | 2.03 ** (1.02–4.05) | 1.36 (0.66–2.85) | N/A | N/A |
| | Benzimidazoles | 16 | 1063 | Ref. | Ref. | | |
| Hepatic disorders | **Ivermectin** | 50 | 523 | 0.61 (0.44–0.86) | 0.51 (0.36–0.74) | 1.36 (0.08–21.9) | 0.50 (0.35–0.73) |
| | Benzimidazoles | 145 | 934 | Ref. | Ref. | Ref. | Ref. |
| Cardiac failure | **Ivermectin** | 6 | 567 | 11.4 ** (1.37–94.9) | N/A | N/A | N/A |
| | Benzimidazoles | 1 | 1078 | Ref. | | | |
| Rhythm disorders | **Ivermectin** | 7 | 566 | 3.32 * (0.97–11.40) | 3.18 * (0.86–11.7) | N/A | 3.18 * (0.86–11.7) |
| | Benzimidazoles | 4 | 1075 | Ref. | Ref. | | Ref. |
| Mazzotti's reaction | **Ivermectin** | 36 | 537 | 2.94 *** (1.74–4.99) | 2.16 ** (1.16–4.03) | 1.95 ** (1.09–3.52) | 19.7 *** (2.20–175.5) |
| | Benzimidazoles | 24 | 1055 | Ref. | Ref. | Ref. | Ref. |

[a] Adjusted for origin (Sub-Saharan Africa or RoW), gender, age and period of notification

[b] Adjusted for gender, age and period of notification

* Drug reaction with eosinophilia and systemic symptoms

** Severe Acute Respiratory Syndrome.

in the world, allowing us to estimate ROR for rare events with sufficient statistical power and to stratify on geographical origin (SSA vs. RoW). Second, analyses were performed with adjustment for several potential confounders such as origin, gender, age and period of notification. Third, nearly all results of our principal analysis were confirmed in our sensitivity analyses considering all antinematodal agents. Fourth, our results are consistent with already known risk associated with ivermectin (encephalopathy in SSA and Mazzotti reactions).

Limitations of this study include the concern about under-reporting of suspected ADRs and differences in the under-reporting between different countries as well as the lack of information on the number of drug administrations, which is a major disadvantage inherent in studies using pharmacovigilance databases [33,34]. Although under-reporting may be less important since we focused on serious ADRs (which are more likely to be reported) [35], our

**Table 7. Possible/Probable *Loa loa* encephalopathy temporally related to Mectizan and other encephalopathies related to ivermectin: mains risk factors, symptoms and mechanisms involved.**

| | Main risk factors | Main symptoms | Main mechanisms involved |
|---|---|---|---|
| **Possible/Probable *Loa loa* encephalopathy temporally related to Mectizan (PLERM)** | - Intensity of the initial *Loa* microfilaremia | - *12-24h following treatment*: fever, fatigue, arthralgia, agitation, mutism, incontinence<br>- *24-72h following treatment*: consciousness disorders including coma and extrapyramidal signs, typical hemorrhages in the palpebral conjunctiva, retinal lesions<br>- Existence of diffuse pathological process at electroencephalogram level | - Paralysis of the microfilariae due to the action of ivermectin resulting in embolisms in the brain capillaries<br>- Inflammatory processes at the cerebral level |
| **Other encephalopathies related to ivermectin:**<br>**- Toxicosis due to an overdose**<br>**- Toxicosis due to a mutation** | - Polymorphism of MDR1 gene<br>- Deficiency in P-glycoproteins<br>- Intentional or unintentional overdosing | - *Few hours after administration*: nausea, vomiting, abdominal pain, salivation, tachycardia, hypotension, ataxia, pyramidal signs, binocular diplopia<br>- Normal paraclinical tests results | - Passage of ivermectin through the blood-brain barrier (due to overdose or mutation of transporters/metabolism actors) |

analyses cannot measure the real risk of ADR but only the differences in reported events. Indeed, subjects in the control group (non-cases) are not healthy controls but patients with other various reported ADRs and pharmacovigilance data do not consider the total amount of patients exposed to the drug. Nevertheless, there is no apparent reason that, in a specific region, ADRs would be more or less reported with ivermectin than those occurring after treatment with benzimidazole drugs or other antinematodal drugs. By analyzing real-life surveillance data, disproportionality analyses have demonstrated their usefulness for detecting drug risks [36,37]. Anyway, these results should be taken with caution because of potential missing information. Pharmacovigilance systems are not yet well established in SSA countries. In 2017, only 30% of these countries had specific procedures for the monitoring of ADRs and only 28% had a platform for coordinating pharmacovigilance activities at the national level [38]. Cases of serious adverse events occurring during the ivermectin mass drug administration organized by the onchocerciasis and LF control programs have to be reported by the countries to the Mectizan Donation Program, but the extent to which all relevant observations are recorded in the rural areas where onchocerciasis and LF are endemic and then passed on to the central level is unknown as is the extent to which they are reported into the WHO VigiBase. For example, we found no cases from Cameroon even though ivermectin mass drug administration programs have been ongoing there for 30 years, and many cases are known to have occurred since the early 1990s [39]. In addition, the first case of a post-ivermectin ADR was reported in VigiBase in December 2003 while the first reported death after ivermectin was reported by the WHO Drug Information in 1991 [40]. We consider it likely that availability of complete data from SSA would show more cases associated with ivermectin use and would increase the strength of the safety signals we identified.

In addition, a notoriety bias (selection bias in which a case has a greater chance of being reported if the drug is known to cause, thought to cause, or likely to cause the event of interest [41]) could be considered for reports of encephalopathy in SSA given that the first cases of encephalopathies involving ivermectin led to complications in the early mass drug administration campaigns for elimination of onchocerciasis. However it is unlikely that such bias exist for two reasons (i) in SSA countries, ivermectin is distributed as part of mass treatment organized by the Ministries of Health, and those of *L. loa*-endemic countries might be less inclined to report post-ivermectin sADRs because the cases are not regarded as exceptional and (ii) we also found a strong disproportionality signal in the RoW which is not being affected by this bias.

Our analyses identified serious ADRs that can be associated with ivermectin use that to date have received little, if any attention. As ivermectin is currently widely used off label, especially in Latin America, to control COVID-19 without strong evidence for beneficial effect [42], this study is timely to describe the various suspected sADRs to which this population is potentially exposed even in the absence of onchocerciasis and loiasis endemicity. While ivermectin's excellent safety profile is the basis for mass drug administration campaigns and progress towards elimination in particular of onchocerciasis, one must remain aware and vigilant about the sADRs it may possibly induce.

## Supporting information

**S1 Table. A.** Most frequently reported serious ADRs for each System Organ Class (SOC). If several sADRs belonging to the same SOC are reported in a single patient (ICSR form), the SOC is counted only once in the total. **B.** Most frequently reported serious ADRs for each System Organ Class (SOC) in SSA. If several sADRs belonging to the same SOC are reported in a single patient (ICSR form), the SOC is counted only once in the total. **C.** Most frequently reported serious ADRs for each System Organ Class (SOC) in RoW. If several sADRs belonging to the same SOC are reported in a single patient (ICSR form), the SOC is counted only once in the total.
(DOCX)

**S2 Table. Disproportionality analysis of serious adverse reactions associated with ivermectin compared to other antinematodal drugs.**
(DOCX)

**S3 Table. STROBE Statement—Checklist of items that should be included in reports of case-control studies.**
(DOC)

## Acknowledgments

The Uppsala Monitoring Centre has provided the data but the study results and conclusions are those of the authors and not necessarily those of the Uppsala Monitoring Centre, National Centers, or WHO.

## Author Contributions

**Conceptualization:** Jérémy T. Campillo, Michel Boussinesq, Jean-Luc Faillie, Cédric B. Chesnais.

**Data curation:** Jean-Luc Faillie.

**Formal analysis:** Jérémy T. Campillo.

**Project administration:** Jean-Luc Faillie, Cédric B. Chesnais.

**Resources:** Jean-Luc Faillie.

**Supervision:** Michel Boussinesq, Sébastien Bertout, Jean-Luc Faillie, Cédric B. Chesnais.

**Validation:** Michel Boussinesq, Sébastien Bertout, Jean-Luc Faillie, Cédric B. Chesnais.

**Visualization:** Jérémy T. Campillo.

**Writing – original draft:** Jérémy T. Campillo.

**Writing – review & editing:** Jérémy T. Campillo, Michel Boussinesq, Sébastien Bertout, Jean-Luc Faillie, Cédric B. Chesnais.

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
