## [Decision Letter · Decision Letter 0]

9 Mar 2021

Dear Dr Campillo,

Thank you very much for submitting your manuscript "Serious adverse reactions associated with ivermectin: a systematic pharmacovigilance study in sub-Saharan Africa and non-sub-Saharan African regions" for consideration at PLOS Neglected Tropical Diseases. As with all papers reviewed by the journal, your manuscript was reviewed by members of the editorial board and by several independent reviewers. The reviewers appreciated the attention to an important topic. Based on the reviews, we are likely to accept this manuscript for publication, providing that you modify the manuscript according to the review recommendations. 

Sincerely,

Samuel Wanji

Associate Editor

Abhay Satoskar

Deputy Editor

Reviewer's Responses to Questions

**Key Review Criteria Required for Acceptance?**

**Methods**

-Are the objectives of the study clearly articulated with a clear testable hypothesis stated?

-Is the study design appropriate to address the stated objectives?

-Is the population clearly described and appropriate for the hypothesis being tested?

-Is the sample size sufficient to ensure adequate power to address the hypothesis being tested?

-Were correct statistical analysis used to support conclusions?

-Are there concerns about ethical or regulatory requirements being met?

Reviewer #1: The objectives are clear, and the design is appropriate for the objectives. The population was clearly described and the whole targeted population was included according to a predefined inclusion criteria. Authors were compliant to ethical and regulatory requirements.

Reviewer #2: (No Response)

Reviewer #3: Very well planned, executed and presented study.

**Results**

-Does the analysis presented match the analysis plan?

-Are the results clearly and completely presented?

-Are the figures (Tables, Images) of sufficient quality for clarity?

Reviewer #1: Results match the analysis plan

Results are completely presented

Figures and tables are of sufficient quality and clarity.

Reviewer #2: Line 259, “table 4” dizziness is duplicated. should be removed

Reviewer #3: Data analysis and result presentation are clear.

**Conclusions**

-Are the conclusions supported by the data presented?

-Are the limitations of analysis clearly described?

-Do the authors discuss how these data can be helpful to advance our understanding of the topic under study?

-Is public health relevance addressed?

Reviewer #1: -Are the conclusions supported by the data presented? YES

-Are the limitations of analysis clearly described? YES

-Do the authors discuss how these data can be helpful to advance our understanding of the topic under study? YES

-Is public health relevance addressed? YES

Reviewer #2: (No Response)

Reviewer #3: conclusions, recommendations, strengths and limitations of the study mentioned.

**Editorial and Data Presentation Modifications?**

Reviewer #1: 1- The title can be rewritten to reflect the study design where ivermectin pharmacovigilance study was compared between sub-Saharan Africa and the rest of the world and not it was not limited to a comparison between "sub-Saharan Africa and non-sub-Saharan African regions"

2- key words: "antinematodal drugs" can be added to the keywords for better search results.

3- Page 4 line 79-83. The paragraph needs rewriting for clarification. This paragraph also needs a reference and adding the route of administration for MDA strategy

4- Page 5 line 102: there is no need for the sentence "see below".

For the methodology and results part, please check the attachment.

Reviewer #2: (No Response)

Reviewer #3: very well presented

**Summary and General Comments**

Reviewer #1: The study will benefit so many people within the healthcare field, as it discusses the adverse events of ivermectin that have been reported during an 18 years period. The disproportionality method used is novel for this drug and its limitations have been clarified and discussed by the authors as well as its strengths. The science is sound and the language is clear. It also highlights the importance of proper pharmacovigilance procedures needed in many different parts of the world including the sub-Saharan Africa as an example. 

I have a few points to be considered by the authors. 

I have included them in the attachment.

Reviewer #2: 1. No comments regarding ethical approval was mentioned. waived ?

2. Line 75 “infection with Wuchereria bancrofti” is an off-label use in North America (FDA) and is labeled for use in only in Europe (EMA), the same in line 77 “human sarcoptic scabies”

This is contradicting with line 74 “Europe and North America” which refers to labeled use in both, also contradicting with line 346. So, from line 74 till line 79 should be reframed. 

3. Abbreviations overuse (probably to reduce word count) which may confuse some readers keeping low clarity of the manuscript.

strengths: Good topic however the need for the above mentioned points. Discussed part is well written.

Reviewer #3: very well palnned and presented study.

 ethical committe permission?

PLOS authors have the option to publish the peer review history of their article (what does this mean?). If published, this will include your full peer review and any attached files.

Reviewer #1: No

Reviewer #2: Yes: Mohamed S Abdelmoneim

Reviewer #3: No

Figure Files:

Data Requirements:

Reproducibility:

References

---

## [Decision Letter · Decision Letter 1]

1 Apr 2021

Dear Dr Campillo,

We are pleased to inform you that your manuscript 'Serious adverse reactions associated with ivermectin: a systematic pharmacovigilance study in sub-Saharan Africa and in the rest of the World' has been provisionally accepted for publication in PLOS Neglected Tropical Diseases.

Best regards,

Samuel Wanji

Associate Editor

Abhay Satoskar

Deputy Editor

Reviewer's Responses to Questions

**Key Review Criteria Required for Acceptance?**

**Methods**

-Are the objectives of the study clearly articulated with a clear testable hypothesis stated?

-Is the study design appropriate to address the stated objectives?

-Is the population clearly described and appropriate for the hypothesis being tested?

-Is the sample size sufficient to ensure adequate power to address the hypothesis being tested?

-Were correct statistical analysis used to support conclusions?

-Are there concerns about ethical or regulatory requirements being met?

Reviewer #1: yes

Reviewer #2: (No Response)

Reviewer #3: addressed all comments

**Results**

-Does the analysis presented match the analysis plan?

-Are the results clearly and completely presented?

-Are the figures (Tables, Images) of sufficient quality for clarity?

Reviewer #1: (No Response)

Reviewer #2: (No Response)

Reviewer #3: addressed all comments

**Conclusions**

-Are the conclusions supported by the data presented?

-Are the limitations of analysis clearly described?

-Do the authors discuss how these data can be helpful to advance our understanding of the topic under study?

-Is public health relevance addressed?

Reviewer #1: (No Response)

Reviewer #2: (No Response)

Reviewer #3: addressed all comments

**Editorial and Data Presentation Modifications?**

Reviewer #1: (No Response)

Reviewer #2: (No Response)

Reviewer #3: (No Response)

**Summary and General Comments**

Reviewer #1: (No Response)

Reviewer #2: previous comments were addressed; however ethical approval status should be mentioned even if it was not required.

you may for example An ICSR is an anonymized report hence it does not require or waived ethical approval.

Reviewer #3: addressed all comments

PLOS authors have the option to publish the peer review history of their article (what does this mean?). If published, this will include your full peer review and any attached files.

Reviewer #1: No

Reviewer #2: **Yes: **Mohamed S Abdelmoneim

Reviewer #3: No

---

## [Editor Report · Acceptance letter]

14 Apr 2021

Dear Dr Campillo,

We are delighted to inform you that your manuscript, "Serious adverse reactions associated with ivermectin: a systematic pharmacovigilance study in sub-Saharan Africa and in the rest of the World," has been formally accepted for publication in PLOS Neglected Tropical Diseases.

Best regards,

Shaden Kamhawi

co-Editor-in-Chief

Paul Brindley

co-Editor-in-Chief
